# Lab-on-a-Chip Electrochemical Immunosensor Array Integrated with Microfluidics: Development and Characterisation

**Shifa Felemban [1], Patricia Vazquez [2], Thanih Balbaied [3] and Eric Moore [3,*]**

1    Department of Chemistry, Umm Al-Qura University, Mecca 24382, Saudi Arabia
2    Smart Sensors Lab, Lambe Institute for Translational Research, National University of Ireland, H91 TK33 Galway, Ireland
3    Sensing & Separation Group, School of Chemistry and Life Science Interface, Tyndall National Institute, University College Cork, T12 R5CP Cork, Ireland
*    Correspondence: e.moore@ucc.ie

**Abstract:** Lab-on-a-chip has recently become an alternative for in situ monitoring for its portability and simple integration with an electrochemical immunoassay. Here, we present an electrochemical cell-on-a-chip configured in a three-electrode system to detect benzo(a)pyrene (BaP) in water. 11-Mercaptoundecanoic acid (MUA), a self-assembled monolayer (SAM), was used to modify a gold chip surface to reduce the randomness of antibody binding. A carboxylic acid group was activated with -ethyl-3-(3-dimethylaminopropyl) (EDC) in combination with N-hyrodsuccinimide (NHS) before antibody immobilisation. The mechanism of the electrochemical reactions on a gold surface and SAM formation were investigated by cyclic voltammetry and contact angle measurements. The data revealed a lower contact angle in the modified chip and a scan rate of 50 mV/s. Through the addition of modification layers and thiol end groups to the SAM, our design allowed the chip surface to became more insulated. All were tested by amperometric detection using the developed Q-sense system. This novel technique detected multiple samples, and completed the analysis reasonably quickly. While the integrated system proved successful in a lab setting, the aim of the research is to use this system for in situ analysis, which can be brought into a water environment to carry out tests with existing processes. In this way, any issues that may arise from an environmental setting can be rectified in an efficient manner.

**Keywords:** microfluidics; lab on a chip; integration system; electrochemical immunoassay; polycyclic aromatic hydrocarbons (PAHs)

## 1. Introduction

Microfluidics is the technique of manipulating fluids in the tiny channels that are used for inserting samples from the macro-environment. It allows the miniaturisation of a variety of devices and facilitates the integration of a lab-on-a-chip with other systems to analyse complex samples with various antigens. It has had a revolutionary effect on, for example, the environmental monitoring of toxic polycyclic aromatic hydrocarbons (PAHs) from pesticides and herbicides and the monitoring of water supplies using a combination of lateral-flow and enzyme-linked immunosorbent assays with enzymatic glucose biosensors. Academic research has also indicated the presence of a variety of smart materials in the immunoassay system [1–3].

Electrochemical immunoassays are important for developing selective sensors that can be integrated in microfluidics, and the demand for them in the U.S. is rising at an annual rate of 6% [4]. Besides their simplicity and fast detection, they are cost-effective due to the use of inexpensive screen-printed electrodes that permit the thorough screening of a large number of samples. The electrochemical signals that occur on the surface are more accurate than from optical detection instruments, which suffer from light scattering and detection limitations. As samples from the environment have a high surface-to-volume

ratio, detection in microfluidics can be affected; therefore, it is necessary to consider the surface properties of the electrode in its design and function. During analysis, the surface can be treated with a coating to prevent non-specific adsorption or an unwanted charge, leading to high output accuracy. The electrochemical signals will depend on the modification and fabrication quality of the electrodes and on the interaction of the receptor with the surface of the working electrodes.

Many attempts have been made to detect PAHs in microfluidics [5] including a covalent organic framework [6], carbon-based magnetic sorbents [7], ionic-liquid-modified magnetic nanoparticles [8] and nanoporous silicon organosilicate [9]. Despite their advantages, selectivity, reproducibility and reusability are still issues because environmental pollutants come in many forms. A chemical immunoassay provides a wide and selective detection of a large number of contaminants for use as antigens, and its result is reliable because it is dependent on enzymatic reaction. In addition, integrating immunoassays in microfluidics results in faster results since the short diffusion lengths allow for easier enzymatic reactions [10,11]. Therefore, real-time, rapid, and accurate on-site detection is a viable strategy for environmental analysis. Since the introduction of these integrated immunoassays, screening platforms have witnessed increasing interest from businesses.

Today, numerous novel detection systems have been used to develop in situ analyses that use a variety of theories for the design of sensitive and fast detection screening platforms, including impedance-based, resonant acoustic-wave-based, calorimetric-based, and micro-electromechanical-based systems [12]. For this paper, we selected an amperometry-based system using a developed Q-sense system: a control module that includes interfaces for microfluidics, immunosensors and user displays (Figure 2). In addition to evaluating various processes simultaneously, Q-sense is a microfluidic-associated system that tracks the generated data [13,14], which was useful for our multi-sensor gold chip. Therefore, an electrochemical immunoassay integrated with multi-sensor detection in microfluidics was applied in the study. The gold chip was modified by a self-assembled monolayer (SAM) of 11-mercaptoundecanoic acid (11-MUA) to facilitate protein immobilisation, and it was characterised by using cyclic voltammetry and contact angle techniques. Real-time amperometry was then used to evaluate the enzymatic reaction targeting benzo(a)pyrene (BaP), which is one of the PAHs. The systems then incorporated microfluidics and were assessed again. These integrated systems improve the performance of traditional methods by taking advantage of microfluidics, which are renowned for their ability to reduce reagent volumes and develop microchannel surfaces.

## 2. Methodology

### 2.1. Reagents

11-MUA, SAM, -ethyl-3-(3-dimethylaminopropyl) carbodiimide (EDC), N-hyrodsuccinimide (NHS), diethanolamine (DEA), potassium chloride (KCl), potassium ferricyanide/ferrocyanide ($K_3[Fe(CN_6)]/K_4[Fe(CN_6 \cdot 3H_2O)]$), sodium hydrogen carbonate ($NaHCO_3$), sodium chloride (NaCl), detergent Tween-20, Tris salt, 1-isopropyl, ethanol, benzo(a)pyrene (BaP), alkaline phosphatase (AP), labelled goat anti-mouse antibodies (whole molecules), all were purchased from Sigma Aldrich (Dublin, Ireland). Other chemicals—1-ethyl-3-(3-dimethylaminopropyl) carbodiimide (EDAC), bovine serum albumin (BSA), and hydrochloric acid (HCl) 37%—were purchased from KMG ultra chemicals ltd (Riddings, UK). Monoclonal mouse antibody 4D5 was bought from Santa Cruz (Heidelberg, Germany), and para-aminophenyl phosphate (pAPP) was purchased from Gold Biotechnology Inc., (Olivette, MO, USA). The solutions were prepared with analytical-grade reagents and with nanopure water (ELGA Lab Water Systems, Kildare, Ireland) when water was required.

### 2.2. Instrumentation

Electrochemical measurements were performed using a PalmSens potentiostat (Palm Instrument BV, (Houten, The Netherlands) and incubation was performed in a Biometra OV3 Incubator at 37 °C (Gottingen, Germany). Removal of organic molecules from the

surface of the chip was performed using a Plasma Cleaner at 45 W (Harrick Plasma, Ithaca, NY, USA). A nitrogen spray gun was used to dry the chip between measurements. The fabricated chip was a three-electrode system (gold as the working and counter electrode and Ag/AgCl reference electrodes) (Figures 1B and 3A. The chip was configured in a multi-electrode array screening for real time environmental measurements as shown in Figure 3B. Design and fabrication of these electrodes was performed at the Tyndall National Institute in Cork, Ireland. Each electrochemical cell had a working electrode (1.822 mm$^2$), counter electrode (3.280 mm$^2$) and reference electrode (2.18 mm$^2$) as seen in Figure 1A. The overall chip size was 6.733 × 24.7 mm.

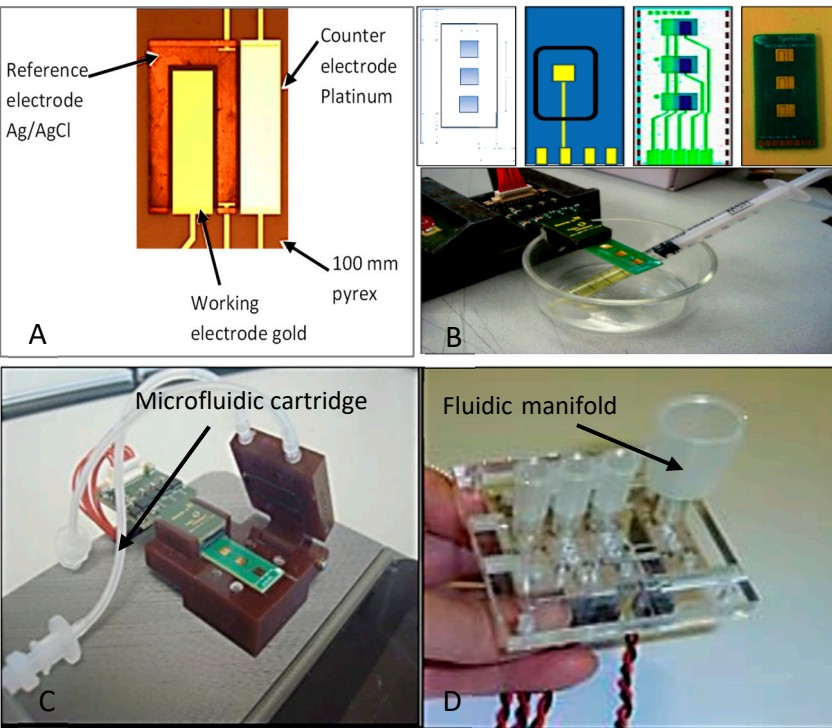

**Figure 1.** (**A**) The three-electrode system (gold as working electrode and platinum as counter electrode and Ag/AgCl reference electrodes). (**B**) Fabricated chip in a three-electrode system. (**C**) Microfluidic cartridge with immunosensor and (**D**) fluidic manifold with valves.

For the design of the fluidic interface, a cartridge was fabricated by injection moulding, as shown in Figure 1C. Its main functions were to hold the fluids under analysis and to cover the surface of the gold chip. The fluid was driven via a set of valves and a portable pump controlled by the electronic module, which was responsible for fluidic control (Figuers 1D and 3C–E) shows the fluidic manifold with valves. Figure 2 illustrates the fluidic channels across the chip as well as the portable pump and the valve control Table Figure 3A shows an image of a euro coin to help clarify dimensions of the setup. An amperometric electrochemical detector embedded in the microfluidic device was used to detect the presence of an electrochemically active species in the sample. Models used in this research were built using COMSOL Multiphysics.

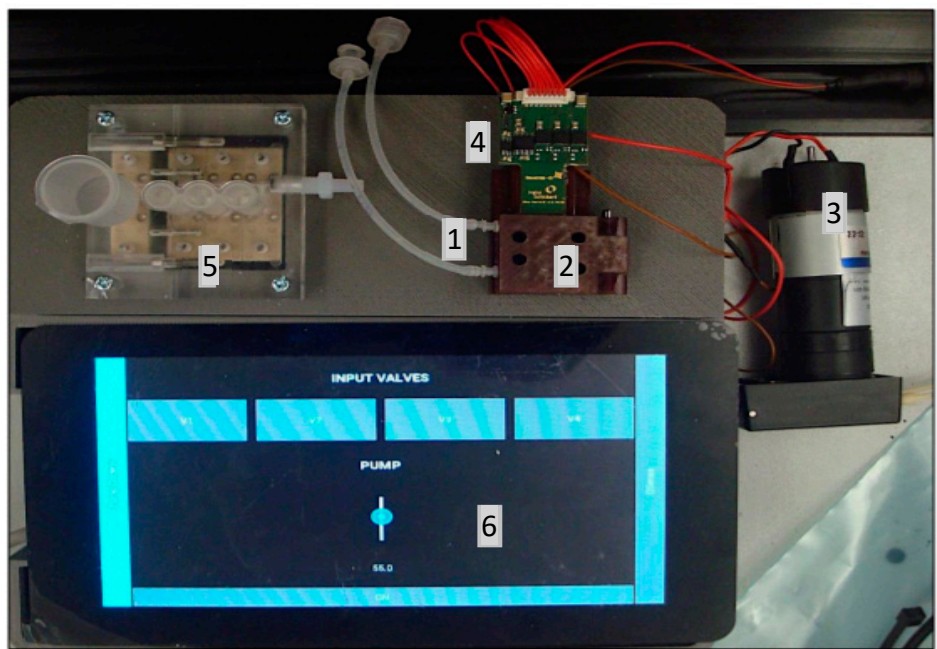

**Figure 2.** Panoramic view of Q-sense with chip integration; (1) fluid channels across the chip, (2) fluidic cells with electrode chip electronics, (3) portable pump, (4) connection to adaptors or electronics, (5) fluidic manifold with valves, and (6) valve control Tab.

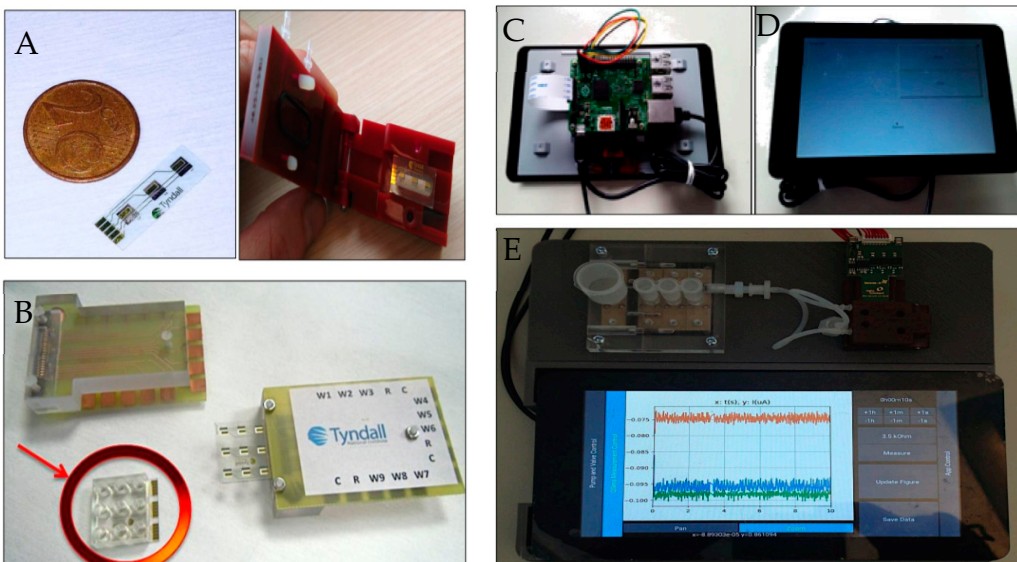

**Figure 3.** (**A**) image of a euro coin to clarify the dimensions of the setup and the microfluidic cell with the chip. (**B**) electrochemical sensing array with interface for connection to PalmSens detector and reservoir device for sample addition. (**C**) control module, containing (**D**) user display and (**E**) interfaces for microfluidics, immunosensor.

*2.3. Preparation of the Gold Chip*

To ensure that the gold chip surfaces were clean for electrochemical tests, they were optimised with two cleaning protocols: ultrasonication and plasma treatment. In the first method, the gold chips were cleaned in acetone and then in 1-isopropyl alcohol and finally in deionised water. Each clean took 5 min. In the second method, the chips were treated with plasma cleaner for 20 min at 45 W. Cyclic voltammetry measurement was used to record the data.

## 2.4. Modification of the Gold Chip

With the 11-MUA SAM, the working electrode of the immunosensor was modified to increase its sensitivity. Activation of water-soluble carbodiimide was performed using EDC and NHS before antibody immobilisation. Because 11-MUA SAM functioned effectively in an ethanolic solution, the antibody bound better to the surface to detect BaP in water samples [15]. Characterisation of the 11-MUA formations on the immunosensor surface was carried out according to the method reported in [16]. The diagram in Figure 4 illustrates the steps of gold chip modification and the measurement used in each step.

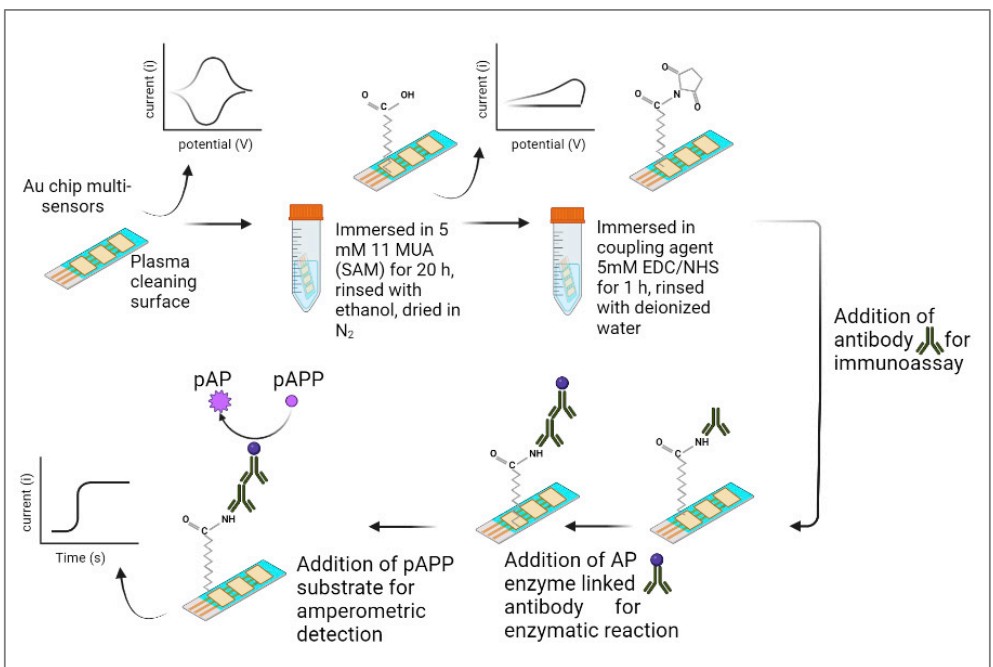

**Figure 4.** Diagram of gold chip modification steps and electrochemical measurements.

## 2.5. Immuonassay

Monoclonal mouse antibody 4D5 specific to BaP (the captured antibody) at the desired concentration was added to the modified gold chip. Then, 1 µL bovine serum albumin–phenanthrene (BSA-PHEN) coating conjugate (6.0 µg/mL) was added to the chip and left to bind at 37 °C for 30 min. A three-step wash with 0.05 M + 0.05% Tris buffer (pH 7.4) and Tween 20 was performed on the gold chip to remove any unbound coating and then dried with the nitrogen spray gun. During the incubation (30 min at 37 °C), the gold chip was blocked by the addition of 4 µL of Tris 0.05 M at natural pH 7.4 to reduce background interferences. The washing step was applied again to remove any unbound materials. Finally, 2 µL of AP-labeled antibody (anti-mouse IgG, 1/3000 dilution) was added and the chip was incubated again for 30 min at 37 °C. For the enzymatic reaction assay, 3 mM of pAPP was added to 0.1 M DEA buffer (9.5), and amperometric detection was conducted.

## 2.6. Cyclic Voltammetry

Electrochemical characterisation measurements were carried out using a PalmSens portable potenionstat. To connect the gold chip to the PalmSens, the interface for the three-electrode system acted as a connector. The PalmSens was connected to a laptop computer to conduct lab tests. The gold chip was tested using cyclic voltammetry (CV), which detects the redox behaviour of species within a potential range. At 50 mV/s in 0.1 M KCl with a ferri/ferrocyanide concentration of 5 mM, CV measurements were carried out in a potential range from −0.2 to 0.8 V. To investigate the scan rate effects, the chip was characterised between −0.2 and 0.8 V in the aqueous electrolyte of ferri/ferrocyanide in

the presence of 0.1 M KCl. The CV measurements were performed at room temperature (25 °C) and at multiple scan rates ranging from 5 to 200 mV/s.

### 2.7. Contact Angle Measurements

Drop contact angle tests were performed to measure the surface hydrophobicity of the SAM surface. To form the droplet, deionised water was pumped out of a syringe system at the medium rate of 1 μL/s, and a 1 μL drop was placed on the surface. Tests were conducted on three chips.

### 2.8. Amperometric Detection

Amperometric measurement with PalmSens allows for a simple device to be used as a portable solution for in situ measurement. The applied potential to detect pAPP was 0.3 V. The measurements were then performed and the current allowed to reach its stable baseline after 60 s. In the buffer, pAPP was dissolved at a concentration of 3 mM. The pAPP solution was freshly made up and stored in a dark area because it is sensitive to light.

## 3. Results and Discussion

### 3.1. Stability of the Gold Chip

Cleaning and substrate preparation are important steps for studying surface phenomena as they improve the surface state for better electrochemical activity. We employed two established protocols to clean the chip for electrode regeneration. Analysing the potential differences between anodic and cathodic peak separation determines chip cleanliness. Figure 5A illustrates the CV after all the cleaning protocols. A single electron transfer between the ferri/ferrocyanide redox couple is theoretically possible at 59 mV, the experimental value that was closest was ΔEp (58 mV), achieved when the CV was performed using the chip treated with the 45 W plasma cleaner. In plasma surface cleaning, contamination is removed using energetic plasma produced from oxygen gas. Compared to other cleaning protocols, this method shows a well-defined peak [17]. For this reason, we used this method as our cleaning protocol for further electrochemical investigation.

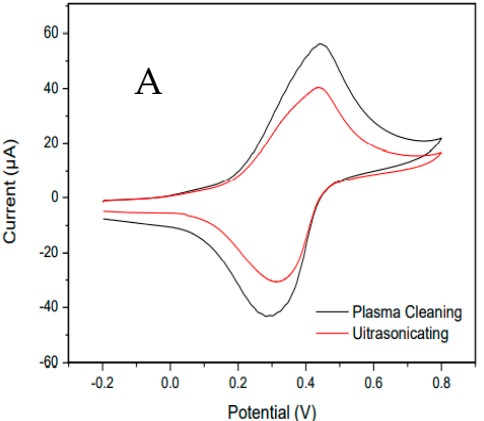 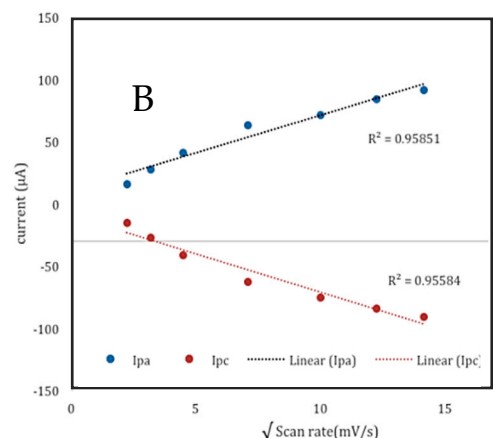

**Figure 5.** (**A**) Studies on different chip cleaning protocols. CV was scanned in 5 mM ferri/ferrocyanide redox couple with a 50 mV/s scan rate. (**B**) Influence of the square root of the scan rates of 5 mM ferri/ferrocyanide on a bare chip. Supporting electrolyte was 0.1 M KCL.

The effect of the scan rate was investigated using CV from 5 to 200 mV/s for the ferrocyanide redox probe. As shown in Figure 5B, a large peak separation was observed when the scan rate increased, resulting in the higher current. It was found that the current was directly proportional to the square root of the scan rate, which indicated that diffusion was limited and governed by the Randles–Sevcik equation. The square root of the applied scan rate correlated well with the anodic peak.

Table 1 below indicates the CV data for the oxidation of a ferri/ferrocyanide concentration of 5 mM with different scan rates. A shift in the anodic peak potential with increasing scan rate indicated a quasi-reversible or "irreversible" reaction.

**Table 1.** Peak potential as a function of scan rate in 5 mM ferrocyanide.

| Scan Rate | $E_{mid}$ vs. Ag/AgCl [a] | $\Delta E_p$ (mV) [b] | $E_{pa}$ (mV/s) [c] |
|:---:|:---:|:---:|:---:|
| 5 | 122 | 228 | 230 |
| 10 | 132 | 256 | 260 |
| 20 | 161 | 318 | 320 |
| 50 | 210 | 340 | 350 |
| 100 | 239 | 282 | 380 |
| 150 | 285 | 290 | 430 |
| 200 | 296 | 288 | 440 |

[a] The values are calculated from the equation of ½ (Epc + Epa) against the Ag/AgCl reference electrode, [b] $\Delta E_p$ = (Epc − Epa), and [c] Ep represents anodic values against the scan rates.

### 3.2. Electrochemical Characterisation of the Modified Chip

To control the orientation of biomolecule (antibody) interaction with the gold chip, 11-MUA self-assembled to modify it. In this approach, the SAM terminated when carboxylic acid was used to modify the gold chip. CV measurements in Figure 6A show a decrease in ferri/ferrocyanide anodic and cathodic peak current after immobilisation of 11-MUA SAM on the chip surface. This situation was caused by the reduced mass transport of highly ordered monolayers that blocked the probe molecules, thus preventing them from penetrating through the well-packed layer. The 11-MUA SAM immobilised on the gold electrode exhibited a low current, indicating surface saturation and the presence of a monolayer of bound molecules. After being terminated by carboxylic acid, the SAM was more effective when EDC/NHS was added. This addition enhanced antibody coverage on the SAM. The primary monoclonal antibody was performed for 30 min at 37 °C after the activation of EDC/NHS. After a protein was introduced, the reaction grew increasingly irreversible, as demonstrated by CV.

The structural information and surface chemistry test of the modified gold surface was determined by using the drop contact angle. Upon comparing the modified gold chip with a bare chip, it was found that the modification on the gold surface influenced the water contact angle. The bare gold chip had a contact angle of 55.07° Figure 6C; for the. modified gold chip, it was lower at 35.58° Figure 6D due to the SAM formation. The low contact angle revealed that the surface of the carboxylic acid monolayer was hydrophilic and consisted of the tail groups of the thiols, which were less hydrophobic.

Amperometric measurements were performed using the PalmSens potentiostat to detect the enzymatic reaction of BaP. A sensing chip with a small working electrode improved the signal to noise ratio and lowered the limit of detection in low concentration. To preserve antibody activity, the chip was coated with layers of biocomponent assays and stored at 4 °C in DEA buffer (pH 9.5). To carry out enzymatic detection, the coated chip was incubated with the AP enzyme and then immersed in a pAPP substrate solution for 60 s with a potential of 300 mV. The baseline current was monitored and allowed to remain stable (see the Figure 6B). At 10 s, 200 μL of pAPP (final concentration 3 mM) was injected into the buffer of the gold chip. The increase in current was recorded as the signal for data evaluation.

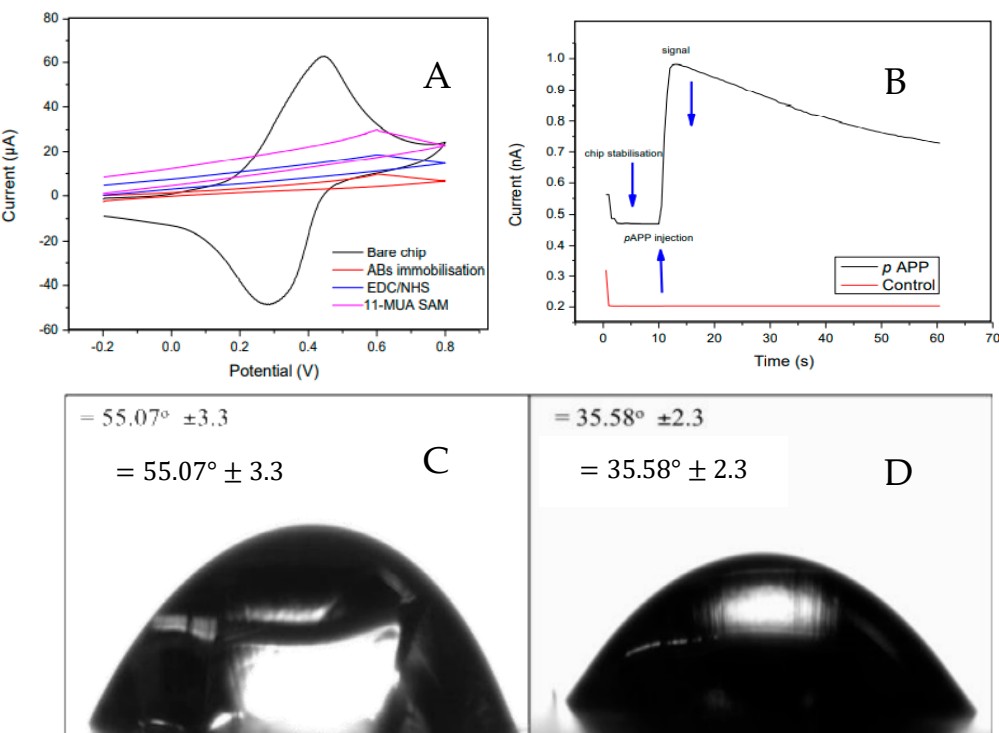

**Figure 6.** (**A**) CV of (i) bare Au: (ii) after applying 11-MUA SAM in 20 h (iii) after addition of EDC/NHS and (iv) after immobilisation of primary monoclonal antibody on surface. The experiment was conducted with an electrolyte containing 5 mM ferri/ferrocyanide redox couple in 0.1 M KCl with a scan rate of 0.05 V/s. (**B**) amperometric detection signal of enzymatic reaction. DEA buffer of pH 9.5 and potential of 300 mV was used as the measurement settings for the modified chip to detect immunoassay of BaP with a 10 s injection of the substrate pAPP. (**C**) Contact angle test before modification and (**D**) Contact angle test after 11-MUA SAM modification on the gold chip.

### 3.3. Evaluation of Microfluidic Channel

In the Q-Sense cell, when the potential was applied to the system, electrostatic force was generated that could be made to cause liquid flows to move, merge, mix, spilt, and dispense from valves; therefore, evaluating the flow of the liquid is important to guarantee flow in the system. Here, in this microfluidic device, simulations were carried out using COMSOL software as seen in Figure 7. The simulation was liquid flow as it came in from the inlet and left at the outlet, which told us the distribution of liquid in the chamber. Blue for high quantity and a lighter colour for low. These simulations provided insight into how the liquid flows in the microfluidic capsule, and in this case the liquid was expected to spread evenly. Visualisation of the flow ensured the avoidance of dead areas, which would not be covered by the liquid. An understanding of the velocity profile in microfluidic channels is important for understanding both how the fluid behaves as it passes through the system and the nature and direction of the fluid particles. As the rainbow goes from blue-to-red, the flow–velocity magnitude distribution corresponds to the intensity differences resulting from cohesive and adhesive forces.

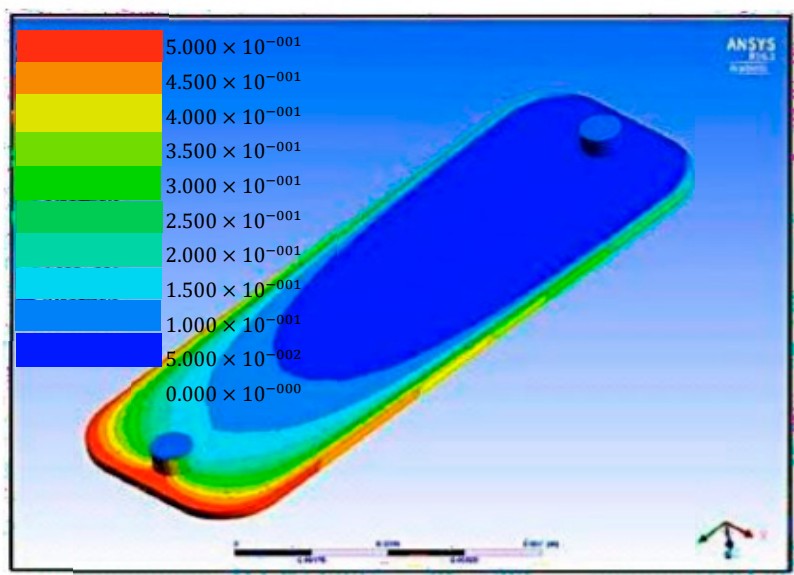

**Figure 7.** COMSOL software that evaluated system flow.

### 3.4. Real-Time Analysis in the Microfluidics Channel

To determine if the developed microfluidic system was compatible with electrochemical measurements, analysis tests were performed to ensure that the system could hold and deliver the sample, and that the electrochemical chip connection worked correctly. The resulting graph, seen in Figure 8A shows the bare chip results from electrochemical detection. The resulting graph also shows that the connections of the system worked efficiently, with no problem in the electrochemical chip or leaking from the sample.

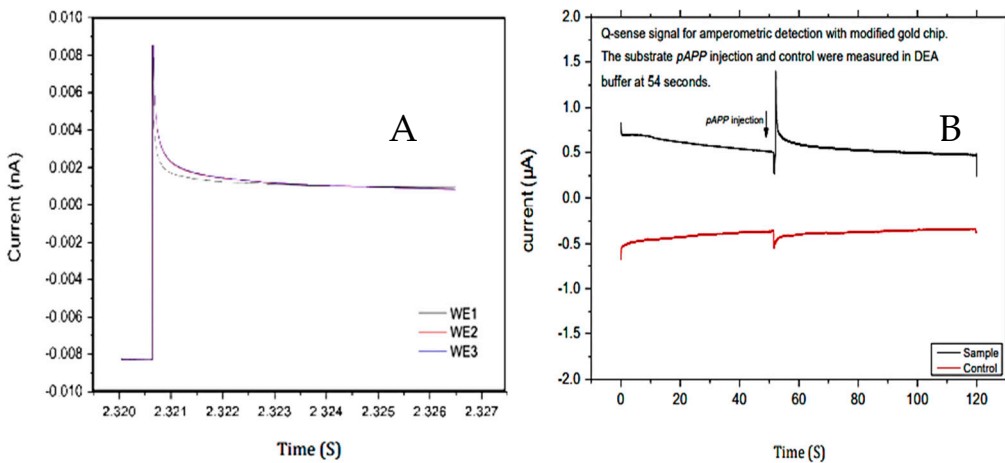

**Figure 8.** (**A**) Electrochemical detection of the bare chip. Electrolyte used was 5 mM ferri/ferrocyanide in 0.1 M KCL. (**B**) Amperometric tests with typical signal of Q-sense. The gold chip array-modified AP bilayer (immunoassay format) was measured in DEA solution.

The graph in Figure 8B indicates the finding of the Q-sense system. On the coated chip, the AP enzyme and pAPP substrate were used for amperometric enzyme detection, which allowed the integration of the sensor for real-time and inexpensive measurement in the field [18]. SAM formation was developed on a chip surface to achieve a specific immobilisation of a captured antibody: monoclonal mouse antibody 4D5. The AP enzyme was used with the pAPP substrate for detection. The captured assay format was used to ensure correct formation on the chip surface (data not shown). The amperometric result showed that the immunoassay complex was efficiently detected.

The results indicated that there was a clear peak for the immunoassay components. At 54 s, the pAPP solution (3 mM final concentration) was delivered to the cartridge chip via the substrate valve, which was controlled by the electronic module. The increase in current was recorded as a signal for data evaluation. The sample with the immunoassay yielded a much higher positive current from amperometric measurements due to the AP concentration in the solution. It also appeared that the current for the immunoassay sample decreased exponentially, while for the bare chip it was almost linear.

## 4. Conclusions

In this paper, we successfully integrated lab-on-a-chip with a microfluidic device configured by the Q-Sense system to detect water pollution. Such a method produced direct and reliable results. The (11-MUA) modification on the gold chip enhanced the immobilisation of proteins on the metallic surface. Data showed irreversible redox reaction, indicating a monolayer of MUA was strongly fixed to the surface. When the antibody was immobilised, the cathodic peak dropped from 30 to 10 µA compared to 60 µA for the bare chip control. It is worth mentioning that the contact angle measurements indicated that the modified chip had a more hydrophilic surface, which positively influenced the stability of proteins on the surface. The affinity assay granted selective detection, and when combined with the electrochemical assay, it showed miniaturisation and real-time detection. The enzymatic reaction assayed by amperometry showed a higher current of 1 µA compared to −0.4 µA for the control. By facilitating the flow of pollutants inside the microchannels, the coated chip allowed portable on-site detection, which also expanded the use of different contaminants according to the type of antigen used. In future research, this system will need in situ tests to address errors and improve the Q-Sense microfluidic system.

**Author Contributions:** Supervision, E.M. and P.V.; Writing—original draft, S.F.; Reviewing—original draft, T.B. All authors have read and agreed to the published version of the manuscript.

**Funding:** This research was funded by the Ministry of Education of Saudi Arabia and Umm Al-Qura University.

**Institutional Review Board Statement:** Not applicable.

**Informed Consent Statement:** Not applicable.

**Acknowledgments:** The authors would like to acknowledge the financial support of the Ministry of Higher Education of Saudi Arabia and Umm Al-Qura University.

**Conflicts of Interest:** The authors declare no conflict of interest.

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
