# Peer review of "Lab-on-a-Chip Electrochemical Immunosensor Array Integrated with Microfluidics: Development and Characterisation"

_2673-3293, doi:10.3390/electrochem3040039_

Round 1

Reviewer 1 Report

The manuscript has been improved. The referee suggests its publication without further revision.

Author Response

Thanks for your comments 

Reviewer 2 Report

The manuscript is a resubmission. It reports a lab-on-a-chip system for PAH detection in environmental water using electrochemical immunosensors integrated with microfluidic channels. The intended application is interesting and the results are promising.

The resubmission includes some changes that address a few of the previous comments. However, other comments remain unaddressed.

- The introduction includes new contents summarizing what has been done in this work.  However, it is still inadequate to serve as rationale for the novelty of this work, i.e. how this work defers from those already reported on eletrochemical immunosensors. There is still no review/citation of existing works that have been done on electrochemical immunoassay and its integration with microfluidics.

- Acronyms are now defined but some still need to be redefined in the main text even if they have already been defined in the abstract.

- Q-sense system needs to be identified with the source supplier and model information if it is commercially obtained, or should be accompanied by a citation of the source.

- Schematic drawings still need to be provided to clearly illustrate the overall integrated structure of the microfluidic cartridge/manifold/valve and the immunosensor. The included photos are far from adequate; the existing drawing is only for the sensor surface. Such drawings can also help understand the functional modules included in the integrated device. 

Author Response

Thanks for your comments

Author Response

Thanks for your comments

Round 2

Reviewer 2 Report

The authors barely addressed my comments. 

For the first comment, it is far from adequate to just say their Q-sense is labelled immunoassay while those in literature is commonly for label-free. Labelled immunoassay with microfluidic devices has been worked on for decades.  Why does Q-sense make their work so special? Not all readers, particularly those outside Europe, even know about Q-sense. 

For 2nd comment, apparently the authors did not bother to properly understand the comment or they inadvertently missed the acronyms that need to be re-defined. An example is "PAH". 

For 3rd comment, refer to above for the first comment. The added citation [12] does help. 

For 4th comment, Figs 1C and 1D, which have no labels and have small sizes and poor quality, may be clear to the authors but are far from enough to provide clear information to readers, who have no prior knowledge of the authors' work. 

I do see the paper shows some interesting work and results, but the overall quality of presentation is not high and the authors seem not willing to make improvements nor to properly understand the suggestions.

Author Response

Dear Journal Editors,

I am enclosing herewith the manuscript entitled “Lab-on-a-chip electrochemical immunosensor array integrated with microfluidics: development and characterisation” for consideration for publication in the Journal of Electrochem

The ID of this article is electrochem-1627512 and here we want to resubmit it back

In this article, we reviewed all comments from all reviewers. Please see all comments below.

comments from 2nd reviewer:

For the first comment, it is far from adequate to just say their Q-sense is labelled immunoassay while those in literature is commonly for label-free. Labelled immunoassay with microfluidic devices has been worked on for decades.  Why does Q-sense make their work so special? Not all readers, particularly those outside Europe, even know about Q-sense. 

It was highlighted blue in manuscript

For 2nd comment, apparently the authors did not bother to properly understand the comment or they inadvertently missed the acronyms that need to be re-defined. An example is "PAH". 

It was highlighted blue in manuscript

For 3rd comment, refer to above for the first comment. The added citation [12] does help. 

Yes, Thank you,

For 4th comment, Figs 1C and 1D, which have no labels and have small sizes and poor quality, may be clear to the authors but are far from enough to provide clear information to readers, who have no prior knowledge of the authors' work. 

Done, see Figure 1 C, 1D and Figure 2

I do see the paper shows some interesting work and results, but the overall quality of presentation is not high and the authors seem not willing to make improvements nor to properly understand the suggestions.

I appreciate your comments, thanks

Reviewer 3 Report

The authors did not address  the following issue:
" A schematic representation of the immunoassay principle should be provided together with a scheme of the AP-catalysed reaction, including the electrode reaction that is responsible for the amperometric detection".

Please check the following papers (they may provide some examples about  the immunoassay principle and the electrode reaction):

1. Fruhmann, P., Sanchis, A., Mayerhuber, L. et al. Immunoassay and amperometric biosensor approaches for the detection of deltamethrin in seawater. Anal Bioanal Chem 410, 5923–5930 (2018). https://doi.org/10.1007/s00216-018-1209-1

2. Heftsi Ragones, David Schreiber, Alexandra Inberg, Olga Berkh, Gábor Kósa, Amihay Freeman, Yosi Shacham-Diamand,
Disposable electrochemical sensor prepared using 3D printing for cell and tissue diagnostics; Sensors and Actuators B: Chemical, 216,
2015, 434-442, https://doi.org/10.1016/j.snb.2015.04.065.

Author Response

Dear Journal Editors,

I am enclosing herewith the manuscript entitled “Lab-on-a-chip electrochemical immunosensor array integrated with microfluidics: development and characterisation” for consideration for publication in the Journal of Electrochem

The ID of this article is electrochem-1627512 and here we want to resubmit it back

In this article, we reviewed all comments from all reviewers. Please see all comments below:

comments from 3rd reviewer:

The authors did not address  the following issue:
" A schematic representation of the immunoassay principle should be provided together with a scheme of the AP-catalysed reaction, including the electrode reaction that is responsible for the amperometric detection".

Please check the following papers (they may provide some examples about  the immunoassay principle and the electrode reaction):

  1. Fruhmann, P., Sanchis, A., Mayerhuber, L. et al.Immunoassay and amperometric biosensor approaches for the detection of deltamethrin in seawater. Anal Bioanal Chem410, 5923–5930 (2018). https://doi.org/10.1007/s00216-018-1209-1
  2. Heftsi Ragones, David Schreiber, Alexandra Inberg, Olga Berkh, Gábor Kósa, Amihay Freeman, Yosi Shacham-Diamand,
    Disposable electrochemical sensor prepared using 3D printing for cell and tissue diagnostics; Sensors and Actuators B: Chemical,216,
    2015, 434-442, https://doi.org/10.1016/j.snb.2015.04.065.

I appreciate your comments, thanks, please, see Figure 4

Round 3

Reviewer 2 Report

All my comments were adequately addressed. 

Author Response

Thanks, the English was edited

Reviewer 3 Report

Minor comments:

Figure 4:

- please check the the chemical structure of 11-MUA: replace "-NH" with "OH"

-  replace the cyclic voltammogram  that corresponds to SAM formation onto electrode surface( the second one in the diagram) with an appropriate one, that demonstrates the insulating properties of the self-assembled monolayer.

Figure 4  caption: replace "the" with "The"

Page 11, row 34 replace "contact angel" with "contact angle"

Author Response

Thanks, the vocabulary and the figure was corrected 
